# Energy-efficient CO$_2$/CO interconversion by homogeneous copper-based molecular catalysts

Somnath Guria [1], Dependu Dolui[1], Chandan Das[1], Santanu Ghorai[1], Vikram Vishal[2,3,4,5], Debabrata Maiti[1,3,4], Goutam Kumar Lahiri[1] & Arnab Dutta [1,3,4,5] ✉

Facile conversion of CO$_2$ to commercially viable carbon feedstocks offer a unique way to adopt a net-zero carbon scenario. Synthetic CO$_2$-reducing catalysts have rarely exhibited energy-efficient and selective CO$_2$ conversion. Here, the carbon monoxide dehydrogenase (CODH) enzyme blueprint is imitated by a molecular copper complex coordinated by redox-active ligands. This strategy has unveiled one of the rarest examples of synthetic molecular complex-driven *reversible CO$_2$ reduction/CO oxidation catalysis* under regulated conditions, a hallmark of natural enzymes. The inclusion of a proton-exchanging amine groups in the periphery of the copper complex provides the leeway to modulate the biases of catalysts toward CO$_2$ reduction and CO oxidation in organic and aqueous media. The detailed spectroelectrochemical analysis confirms the synchronous participation of copper and redox-active ligands along with the peripheral amines during this energy-efficient *CO$_2$ reduction/CO oxidation*. This finding can be vital in abating the carbon footprint-free in multiple industrial processes.

Accelerated industrial development to meet the growing energy demands of our modern society has prompted the consumption of fossil fuels at an unprecedented scale resulting in a sharp rise in atmospheric CO$_2$ concentration in recent times. Such a scenario has triggered significant climate change and biodiversity disruption[1–5]. Substantial reduction in CO$_2$ emissions along with rapid decarbonization of industries by switching to renewable energy resources has emerged as primary pathways to abate CO$_2$ emissions and achieve a net-zero state within a reasonable timeframe. A synchronous CO$_2$ capture, utilization, and sequestration method (CCUS) is an indispensable pathway to meet such an ambitious target[6]. In this context, catalytic reduction of CO$_2$ molecules to comparatively active CO can be a critical step in implementing a proactive CO$_2$ management system. CO, a reactive chemical typically generated from partial oxidation

of coke, is one of the highly sought vital reagents for large-scale industrial processes such as the Fischer-Tropsch synthesis of hydrocarbons, the Monsanto/Cativa acetic acid synthesis, iron oxide reduction in blast furnaces, and methanol synthesis from syngas[7–9]. Hence, developing sustainable technology for facile CO$_2$ to CO conversion can unlock new opportunities in the CCUS vertical of decarbonization while paving a sustainable CO$_2$ utilization pathway. Typically, CO$_2$ reduction reaction (CO2RR) leading to CO is an energy-demanding process owing to the inherent structural stability of the CO$_2$ molecule[10,11]. Nature has evolved metalloenzyme carbon monoxide dehydrogenase (CODH), where the synergistic interaction between the active site metal pairs (Ni/Fe) and the presence of the dynamic protein scaffold plays a vital role in exhibiting reversible $CO_2 \rightleftharpoons CO$ conversion. (Fig. 1A)[12,13].

[1]Chemistry Department, Indian Institute of Technology Bombay, Powai, Mumbai 400076, India. [2]Earth Sciences Department, Indian Institute of Technology Bombay, Powai, Mumbai 400076, India. [3]Interdisciplinary Program in Climate Studies, Indian Institute of Technology Bombay, Powai, Mumbai 400076, India. [4]National Center of Excellence in Carbon Capture and Utilization, Indian Institute of Technology Bombay, Powai, Mumbai 400076, India. [5]UrjanovaC Private Limited, Powai, Mumbai 400076, India. ✉e-mail: Arnab.Dutta@iitb.ac.in

Several genres of bimetallic molecular catalysts were developed, imitating the architecture of binuclear CODH for facile $CO_2$ reduction reaction (CO2RR) catalysis[10,14-17]. Among them, the dimers of Ni-cyclam, pyridine-bound, and cryptand-coordinated cobalt complexes displayed early signatures of superior CO2RR activity compared to their monomeric analogues[18-23]. Other variations of functional models were also developed in an attempt to reduce the energy barrier by fine-tuning the mode of $CO_2$ binding, electron transfer to bound $CO_2$, and subsequent activation of $CO_2$ via the tactical modifications of the molecular framework[14,24,25]. Naruta and co-workers have developed a dimer of Fe-based porphyrins for electrocatalytic CO2RR with improved reactivity. However, the presence of bulky aromatic substituents in the ligand framework impeded the use of this active dimer beyond the organic solvents[26]. The generation of substantial amounts of $H_2$ along with CO as CO2RR product, was also a concern for deploying the Fe-porphyrin dimer. Hence, the overall growth in CODH-inspired functional synthetic catalyst design remains in its early stage.

Here in this work, we have probed a set of two copper-based molecular complexes [bis-(2-(phenylazo)pyridine)[PAP] copper(I) perchlorate (**C1**) and bis-(6-amino-2(phenylazo)pyridine) [APAP] copper(I) bromide (**C2**)] for their potential role in $CO_2$ reduction in homogenous conditions (Fig. 1B–D). **C1** was synthesized earlier by Chakraborty and co-workers; however, they did not explore the electrocatalytic properties of the complex[27]. Interestingly, **C1** displayed nearly reversible $CO_2$ reduction/CO oxidation behavior in organic solvent, and reversibility in aqueous solution, albeit at a relatively slower rate (TOF ~ 24 s⁻¹). On the other hand, **C2** exhibited a bidirectional catalytic response in organic solvent with a fast $CO_2$ reduction (TOF ~ 85 s⁻¹) but a distinctly separated slow CO oxidation signal (TOF ~ 3 s⁻¹). The inclusion of water in the solvent gradually improves CO oxidation, and it achieves the reversible $CO_2$ reduction/CO oxidation signature in a 1:1 $CO_2$/CO mixture in 100% aqueous media (pH 7.0

buffer). The presence of the redox-active ligand scaffold is key to achieving such reversible catalysis, which is classically a hallmark of metalloenzymes[28-31]. However, the difference between the activity of **C1** and **C2** can be attributed to the critical influence of the pendant amine group in the ligand scaffold of **C2**. The rational deployment of a series of complementary spectroscopic techniques has allowed us to comprehend the possible electrocatalytic cycle where the copper center and redox-active ligand motif both exchange one electron each during the interconversion of $CO_2$ and CO. These complexes also catalyze $CO_2$ reduction and CO oxidation reactions under chemical conditions in the presence of appropriate sacrificial electron donor (ascorbic acid) or electron acceptor (cerium ammonium nitrate), respectively.

## Results

Both copper complexes are readily soluble in a range of organic solvents (methanol, N, N'-dimethylformamide (DMF), acetonitrile, acetone, dimethyl sulfoxide, ethanol) as well as in an aqueous medium. The preliminary optical study of the complexes was executed in DMF, where three distinct absorbance bands were observed in the region between 300 nm to 900 nm. For both complexes, a strong absorbance feature is present in the near-UV region that is attributed to the ligand-centered π-π* and n-π* transitions[32]. This signal is also present in the solitary ligand solution, and it is red-shifted following the metalation (Fig. S1). On the other hand, a set of two relatively weaker broad absorbance features appeared in the visible region around 580 nm and 700 nm, respectively (Fig. S1, Table 1), which are assigned to metal [$d^{10}$ Cu(I)] to ligand (π*) charge transfer (MLCT). These MLCT bands split up, possibly due to the bifurcation of Cu(I)-based $d$-orbitals into $e$ and $t_2$ symmetries in a tetrahedral coordination geometry[27,33]. These absorbance features retain their trend even in aqueous solutions (Fig. S2, Table 1). The appearance of the characteristic IR stretching

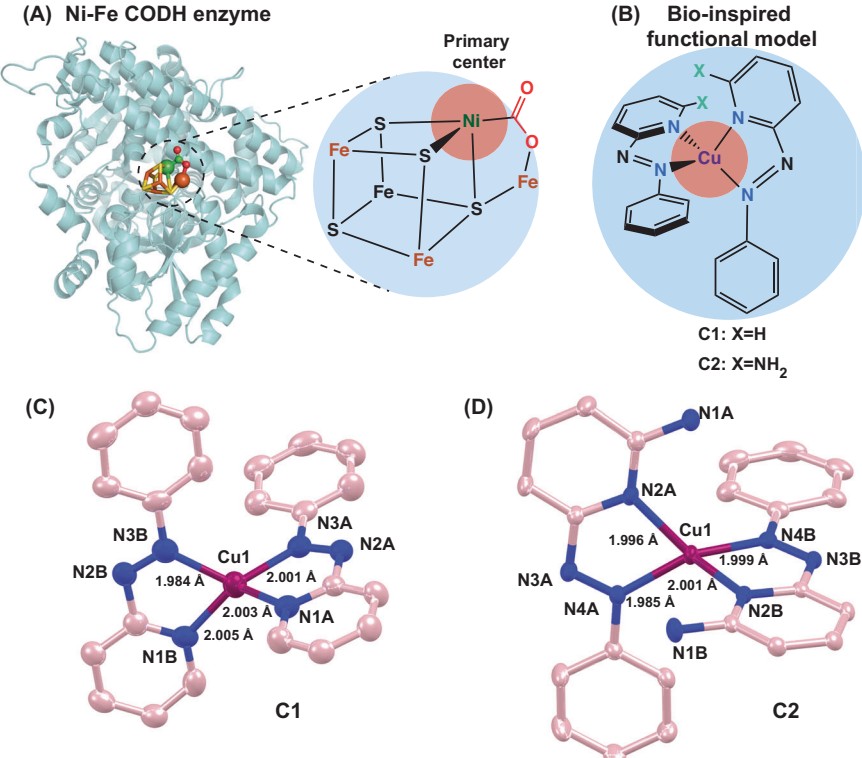

**Fig. 1 | Bio-inspired catalyst design and the crustal structures. A** The structure of Ni-Fe CODH enzyme along with its active site. **B** The generalized chemical structures for complexes **C1** and **C2**. Here, the beige and blue circles depict the central metal center and redox partners, respectively. ORTEP drawings of single crystal structures of **C** complex **C1**, and **(D)** **C2**. Displacement ellipsoids are drawn at the 50% probability level. C, Cu and N atoms are displayed as pink, purple, and blue spheres of arbitrary radii, respectively. Hydrogen and counter ions are omitted for clarity.

**Table 1 | Summary of the fundamental properties of C1 and C2**

| Complex | Optical absorbance [λ/nm (ε in M⁻¹cm⁻¹)] (solvent) | FTIR spectra[a,b] [Band/cm⁻¹ (assignment)] | Reduction potential (V vs. Fc⁺/Fc) | EPR for Cu(II)-complexes | |
|---|---|---|---|---|---|
| | | | | $g_\perp$ | $g_\parallel$ |
| **C1** | 360 (25000), 580 (3929), 700 (907) (DMF) 345 (28000), 577(3548), 700(905) (Water) | 1629 ($v_{C=C/C=N}$) (s) 1594 ($v_{C=C/C=N}$) (s) 1380 ($v_{N=N}$) (m) 1110 ($v_{ClO_4^-}$) (s) | -0.26 (Cu$^{II/I}$), –1.05 (L)[c], –1.45 (L), –1.65 (L), –1.95 (L) | 2.06 | 2.27 |
| **C2** | 425 (26500), 601(5860), 739 (1233) (DMF) 425 (27500), 583 (6448), 738(1699) (Water) | 3440 ($v_{N-H}$) (s) 3255 ($v_{N-H}$) (s) 1625($v_{C=N}$) (s) 1482 ($v_{C=C/C=N}$) (s)1452 ($v_{C=C/C=N}$) (s) 1363 ($v_{N=N}$) (m) | –0.05 (Cu$^{II/I}$), –1.01 (L), –1.33 (L), –1.80 (L) | 2.07 | 2.28 |

[a] Recorded in KBr Pallet.
[b] s strong; *m* medium.
[c] *L* ligand.

signals originating from azo (-N = N-) and pyridinyl motif, along with the $ClO_4^-$ counter anion, further corroborated the formation of the metal complexes (Fig. S3, Table 1)[27,34]. The presence of diamagnetic Cu(I) center in the initial forms of complexes **C1** and **C2** was further supported by EPR studies, where both of them were found to be EPR silent (Fig. S4).

These complexes were oxidized chemically with an equivalent amount of the cerium ammonium nitrate (CAN), generating the corresponding Cu(II) species. The Cu(II)-centric forms of **C1** and **C2** were probed via EPR spectroscopy, where both of them exhibited an axially symmetric signal ($g_\parallel$ ~ 2.27-2.28; $g_\perp$ ~ 2.06-2.07) (Fig. S4, Table 1). Such an EPR signature indicates the presence of a tetrahedrally distorted Cu(II) center.

**Electrochemical results**

Initially, the electrochemical properties of complex **C1** were probed in a dry DMF medium under an argon (Ar) environment. All the potential values in organic media are reported against Fc⁺/⁰ scale. The complex exhibited five reversible electrochemical signatures at −0.26 V, −1.05 V, −1.45 V, −1.65 V and, −1.95 V (vs. $FeCp_2^{+/0}$) (Fig. 2A, B). The linear scan rate dependence of these signals validated the stoichiometric character of each signal (Fig. S5). A complementary spectro-electrochemistry experiment was performed further to probe the molecular identity of these stochiometric features. The Cu(I) signature absorbance band ~580 nm disappeared following an applied potential in the anodic direction ( + 0.06 V vs. $FeCp_2^{+/0}$), which re-emerged once a reductive potential (−0.20 V vs. $FeCp_2^{+/0}$) was implemented (Fig. 2C, Fig. S6). Thus, this signal is ascribed as a Cu(II/I) reduction. This Cu(I)-based MLCT band remained unaffected during the spectro-electrochemical experiment when a potential of −1.15 V was applied (Fig. S7A), indicating a ligand-based reduction. The MLCT signals at 580 nm and 740 nm displayed a drastic shift to 450 nm and 590 nm, respectively, as the solution was kept under further reducing conditions (at −1.55 V vs. $FeCp_2^{+/0}$) (Fig. S7B). Such a blue-shifted MLCT band depicts further filling of ligand-based π* orbitals during this third reduction. The optical absorbance bands remained static for the next two reductions at −1.80 V and −2.05 V (vs. $FeCp_2^{+/0}$) (Fig. S7C, D). The characteristic N = N stretching frequency originated from the azo-motif of the ligand scaffold[27], disappeared continuously around those four cathodic potentials (beyond −1.15 V vs. $FeCp_2^{+/0}$) during a spectroelectrochemical IR experiment (Fig. 2D). This observation further corroborates the assignment of four ligand-based reduction processes following the metal-based reductions during a cathodic scan.

The electrochemical signals of **C1** display significant changes when they are exposed to the $CO_2$ atmosphere at the same potential range. A sharp increase in the current was observed for **C1** during a reductive scan under $CO_2$ starting at −1.25 V, followed by the consistent Cu(II/I) and the initial ligand-based reduction signals, which was evident from both low and high scan rate data set (Fig. 2A, B). No significant reductive feature was observed for the rest of the reductive

scan. During the returning oxidative scan, a sharp peak emerged again ~ −1.25 V, followed by Cu(I/II) oxidation signature at −0.25 V. This new set of signals is found to be catalytic in nature as per the trend of their amplitude change with varying scan rates (Figs. S8, 9). We have assigned these signals as $CO_2$ reduction and CO oxidation features, respectively, where the catalysis occurs with minimal overpotential requirements in either direction. To further probe the $CO_2$/CO catalysis, the cyclic voltammogram of **C1** was recorded in the anodic direction from the onset potential of CO oxidation under 1 atm $CO_2$. The resting potential was held at −0.9 V (where $CO_2$ reduction occurs) for varying times during this experiment. The magnitude of the CO oxidation signal enhanced with increasing equilibrium time indicates the formation of CO during the reduction process (Fig. S10A). The intensity of the catalytic $CO_2$ reduction signal also varied with the amount of $CO_2$ present in the solution for **C1** (Fig. S11). The nearly reversible $CO_2$ reduction and CO oxidation signatures were observed better at low scan rates with a relatively narrower non-Faradaic signal (Fig. 3A, Fig. S12A). Next, the catalytic behavior of **C1** was probed in CO/$CO_2$ blended environment in DMF to have a better insight into its $CO_2$ reduction and CO oxidation signals and analyzed the electrochemical signals in the context of reversibility. Costentin and Artero developed an analogous method to unravel the reversible catalytic behavior showcased by *DuBois*-type catalysts[30,35,36]. Here, different concentration variants of **C1** were prepared in DMF, and their current response was recorded under a 1:1 CO/$CO_2$ gas mixture. Here, both the cathodic $CO_2$ reduction signal and the anodic CO oxidation signal were recorded in a single run. The scan started at the equilibrium potential, and the initial scan direction was towards the anodic direction. Here, the background current response was subtracted from the recorded data beyond the potential where the catalytic current response was noticed (Fig. S13). As shown in Fig. 4, the current response continued to grow beyond the equilibrium potential in either redox direction. Hence, it can be concluded that **C1** displayed nearly reversible $CO_2$ reduction/ CO oxidation behavior in organic media under a 1:1 CO/$CO_2$ atmosphere. This data was recorded for complex **C1** at variable concentrations (Fig. 4A). Here, we have observed that the catalytic $CO_2$ reduction and CO oxidation signals are directedly dependent on catalyst concentration. At lower concentrations, the current responses remain low; however, the distinct signature of both $CO_2$ reduction and CO oxidation are visible on either side of the equilibrium potential (Insert of Fig. 4A). Here, the scan rate was kept constant (50 mVs⁻¹) for recording each data. Next, a series of data was recorded for the same concentration of complex **C1** (1.0 mM) at a variable scan rate. Here, the current responses varied with altering scan rates. However, the background corrected data showcase the presence of both $CO_2$ reduction and CO oxidation at all conditions (Fig. 4B). At low scan rates, the catalyst showcases bias towards $CO_2$ reduction compared to CO oxidation.

As $CO_2$/CO reduction involves proton-coupled-electron-transfer (PCET) steps, the rational inclusion of protic functionalities in the

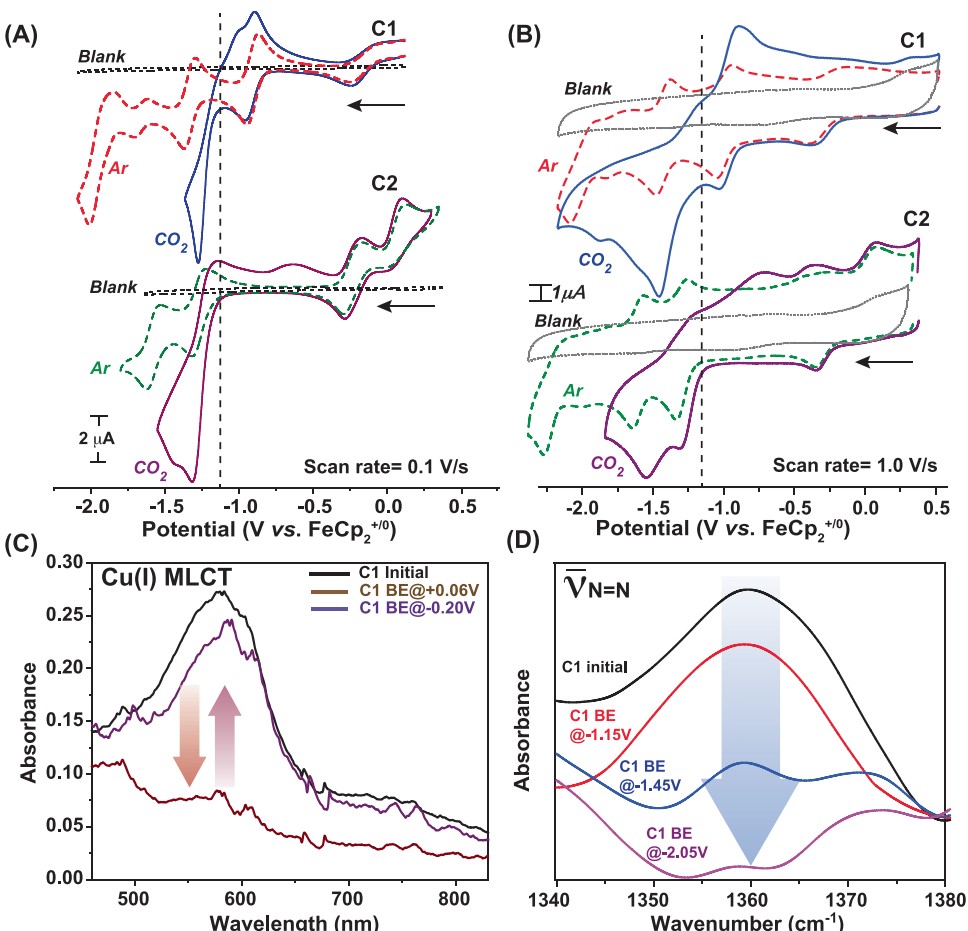

**Fig. 2 | Cyclic voltammetry and spectroelectrochemical studies.** The comparative cyclic voltammograms of 1 mM **C1** (red dotted trace) and **C2** (green dotted trace) along with the blank (dotted grey trace) under Ar atmosphere recorded in DMF media in the presence of $nBu_4N^+BF_4^-$ electrolyte recorded at **A** 0.1 V/s and **B** 1.0 V/s scan rate, respectively. The changes in the electrochemical behavior under $CO_2$ are also included for **C1** (solid blue trace) and **C2** (solid purple trace). Scan rate for each experiment was 1.0 V/s. The horizontal arrows in (**A**) and (**B**) signify the initial scan direction. **C** The optical spectral change of **C1** during spectro-electrochemical experiment when −0.20 V (violet trace) and +0.06 V (brown trace) potential was applied compared to the initial sample (black trace). **D** The serial changes in azo group (·N = N·) stretching band for **C1** during a spectroelectrochemistry-IR experiment with initial sample (black trace) and with the application of −1.15 V (red trace), −1.45 V (blue trace), and −2.05 V (violet trace) under Ar atmosphere.

surrounding can further tune the catalytic behavior[24]. Hence, variable amount of water was added in the DMF solution containing **C1** under $CO_2$ to probe this hypothesis. As shown in Fig. S14, the presence of water shifted the catalytic peaks to the anodic direction as the $CO_2$ reduction signature is now noticed at −0.9 V next to the Cu(II/I) reduction. Next, a series of bulk electrolysis experiments was performed further to establish the formation of CO during $CO_2$ reduction as well as $CO_2$ production following CO oxidation via gas chromatography (GC) experiments (Figs. S15, 16). The bulk electrolysis experiment was performed in variable applied potential covering the onset potential (−0.8 V vs. $FeCp_2^{+/0}$), mid-point potential of $CO_2$ reduction response (−0.95 V vs. $FeCp_2^{+/0}$), and at the maxima of $CO_2$ reduction current (−1.1 V vs. $FeCp_2^{+/0}$) for **C1** in DMF. Here, the maximum charge accumulation and current response were noticed at −1.1 V vs. $FeCp_2^{+/0}$ (Fig. S17). The corresponding GC analysis showcased the evolution of only CO during the catalysis, where the Faradaic Efficiency increased with increasing cathodic applied potential (Fig. S18A). The corresponding experiment on the CO oxidation side displayed the formation of $CO_2$ with an impressive Faradaic efficiency of ~90% (Figs. S18C, S19).

The reversible $CO_2$ reduction/CO oxidation catalysis by **C1** was explored next in aqueous media. Initially, the electrochemical responses of **C1** were recorded in pH 6.5 buffered (0.5 M bicarbonate/

0.5 M phosphate) solution where the respective Cu(II/I) and ligand-based stoichiometric redox signals were noticed (Fig. 3C). The reversible $CO_2$ reduction and CO oxidation catalytic peaks appeared once the solution was exposed to $CO_2$ (Fig. 3C). The reversible feature was specifically noticed when the CV of **C1** was recorded with a low scan rate at a short potential region (−0.65 to −1.1 V vs. SHE) around the $CO_2$/CO equilibrium potential ($E_{CO_2/CO}^0 = -0.53\,V\,vs.\,SHE$) in 1:1 $CO_2$/CO atmosphere (Fig. 3E). **C1**-catalyzed CO generation (via $CO_2$ reduction) and $CO_2$ formation (via CO oxidation) in an aqueous solution was corroborated via corresponding GC experiments following bulk electrolysis (Figs. S20, 21). Thus, the unsubstituted (2-(phenylazo)pyridine) ligand-coordinated complex **C1** showcased a nearly reversible $CO_2$ reduction/CO oxidation behavior persistently in organic, water-blended organic media. Interestingly, **C1** demonstrated a reversible $CO_2$/CO interconversion signature in the aqueous solution.

Next, **C2**, coordinated by the 6-amino-2(phenylazo)pyridine)-substituted APAP ligand, was probed via electrochemistry. The cyclic voltammogram of pendant amine-functionalized complex **C2** under 1 atm Ar demonstrated a sequence of metal and ligand-based four consecutive stoichiometric redox signatures analogous to **C1** (Fig. 2B and S22). However, the Cu(II/I) reduction illustrates an anodic shift in **C2** in comparison to complex **C1** (Fig. S23, Table 1). Additionally, the ligand-based reduction features are also observed at relatively

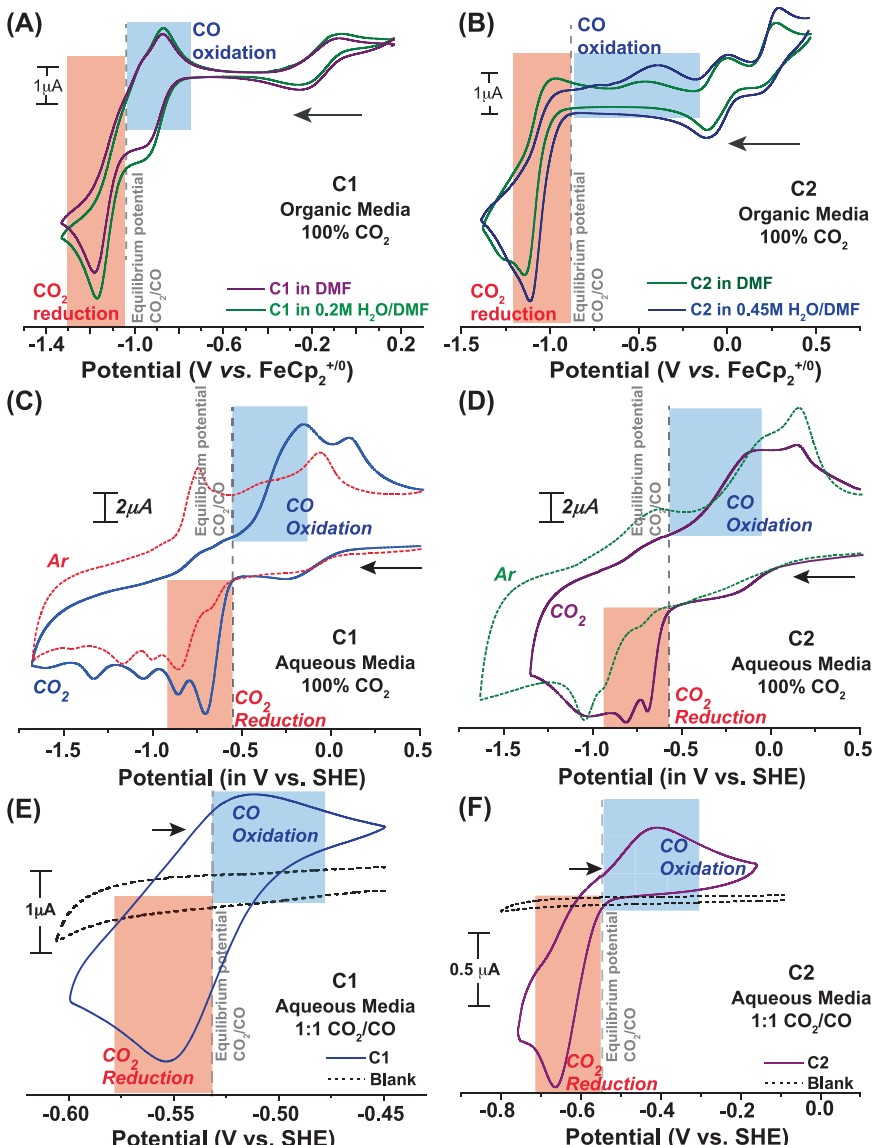

**Fig. 3 | Monitoring CO₂ reduction/CO oxidation interconversion via electrochemistry. A** The comparative cyclic voltammograms of 1 mM **C1** recorded under 1 atmospheric (atm) CO₂ in DMF (purple trace) and 0.2 M water containing DMF (green trace). **B** The analogous data measured for 1 mM **C2** recorded in DMF (green trace) and 0.45 M water containing DMF (blue trace). The scan rate for these experiments were 0.1 V/s and nBu₄N⁺BF₄⁻ was used as electrolyte. The comparative cyclic voltammograms of **C** 0.5 mM **C1** under 1 atm Ar (red dotted trace) and 1 atm CO₂ (solid blue trace), and **D** 0.5 mM **C2** under 1 atm Ar (green dotted trace) and

1 atm CO₂ (solid purple trace) in an aqueous solution of pH 6.5. The scan rate for these experiments were 1.0 V/s and Na₂SO₄ was used as electrolyte. The cyclic voltammograms of **E** 0.5 mM **C1** (solid blue trace) and **F** 0.5 mM **C2** (solid purple trace) under 1 atm of 1:1 CO₂/CO mixture. The blank data under analogous conditions are shown in black dotted trace. The scan rate for these experiments were 0.1 V/s and Na₂SO₄ was used as electrolyte. The horizontal arrows in (**A–F**) signify the initial scan direction. The electrocatalytic CO₂ reduction and CO oxidation responses are represented by the blue and red boxes, respectively.

cathodic direction compared to **C1** (Fig. 2A, B). This change indicates relatively stabilized LUMOs in **C2**, probably owing to better conjugation in (phenylazo)pyridine ligand framework in the presence of the amine group. The red-shifted MLCT signals also corroborate the stabilization of LUMO in **C2** in optical spectral data (Figs. S1, S2). **C2** also exhibits an irreversible oxidation wave at 0.19 V (vs. FeCp₂⁺/⁰), which has been attributed to the oxidation of the bromide counter anion (Fig. S24).

The exposure of **C2** to CO₂ significantly altered the current response as a strong reductive current was noticed during the cathodic scan beyond −1.2 V (vs. FeCp₂⁺/⁰) in DMF media (Fig. 2A, B). Here, the CO₂ reduction signal coincides with the first ligand-based reduction feature observed under Ar. The catalytic nature of this signature was established from the trend of this reductive response under variable

scan rates (Fig. S25). The bulk electrolysis of **C2** performed at this reductive feature in DMF confirmed CO₂ reduction to CO when the experiment was coupled with gas chromatography in a closed container (Fig. S26). An initial cathodic scan (from 0.4 V to −1.5 V) was applied on **C2** in CO₂-saturated DMF, followed by an anodic scan in the opposite direction, where an oxidative feature was noticed. The scan rate dependence data highlighted its catalytic origin (Fig. S27), while this signal was directly influenced by the amount of CO₂ present in the solution (Fig. S28). Hence, this signal was assigned as CO oxidation, which was unambiguously supported by bulk electrolysis data recorded for CO₂ under analogous conditions (Fig. S29). The bulk electrolysis for **C2** was also performed at variable potential, where the maximum current response, charge accumulation, and Faradaic efficiency of CO evolution were noticed at the maxima of the CO₂

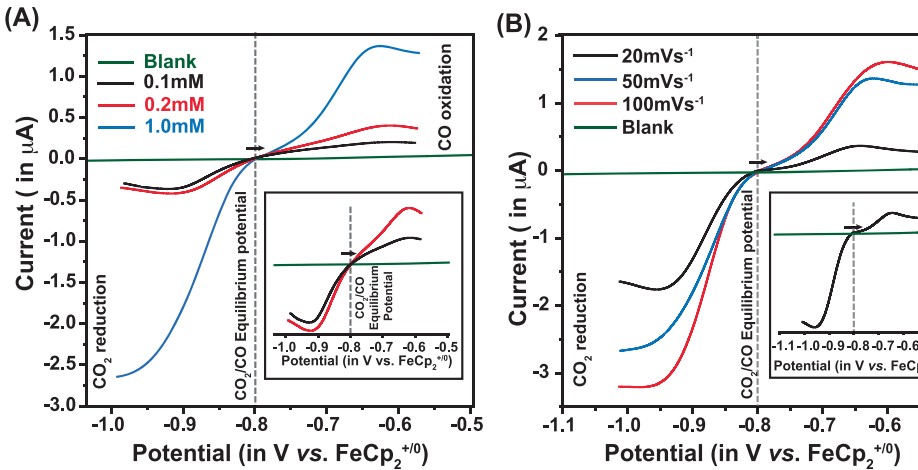

**Fig. 4 | Reversible CO₂ reduction/CO oxidation interconversion. A** The background corrected data highlighting $CO_2$ reduction and CO oxidation signals for 0.1 mM (black trace), 0.5 mM (red trace), and 1.0 mM (blue trace) **C1** in DMF media along with the blank (green trace) under 1:1 $CO_2$/CO atmosphere. Here scan rate was 50 mVs⁻¹. The inset figure demonstrates the data for 0.1 mM (black trace) and 0.5 mM (red trace) **C1** in DMF media along with the blank (green trace). **B** The background corrected data highlighting $CO_2$ reduction and CO oxidation signals for 1.0 mM **C1** recorded at the scan rate of 20 mVs⁻¹ (black trace), 50 mVs⁻¹ (blue trace), and 100 mVs⁻¹ (red trace) in DMF media along with the blank (green trace). The inset figure highlights the 20 mVs⁻¹ (black trace) data recorded in DMF media along with the blank (green trace). Temperature was kept at 300 K for all the experiments.

reduction signal (Fig. S18, Fig. S30). Similar behavior was also observed for the CO oxidation side for **C2**, where the overall CO oxidation was weaker compared to **C1** (Figs. S18, S19).

Interestingly, the oxidative CO oxidation feature was significantly subdued while the peak maximum was shifted anodically from the $CO_2$/CO equilibrium potential ($E^0_{CO_2/CO}$) while the $CO_2$ reduction onset potential aligned to it. Such a behavior can be assigned as a bidirectional $CO_2$ reduction/CO oxidation, which is starkly different from the nearly reversible $CO_2$ reduction/CO oxidation signature observed for **C1** (Fig. 3A, B). This observation indicates that **C2** is biased towards reduction over oxidation during $CO_2$/CO interconversion[37]. However, the **C2**-driven CO oxidation can be modulated with respect to the onset potential and catalytic maxima location with the addition of water in the reaction media (Fig. 3B). This data potentially indicates the key role of protic solvent water during the catalysis, possibly through the interaction with the peripheral amine groups. This hypothesis was corroborated by the altered $CO_2$ reduction and CO oxidation features for **C2** in $CO_2$-saturated DMF solution in the presence of $H_2O$ vs. $D_2O$ (Fig. S31).

The influence of protic solvent on the **C2** electrochemical behavior was evident when its catalytic activity was monitored in an aqueous solution (pH 6.5) under 1 atm $CO_2$. The CO oxidation signal intensity was improved with a further shift toward $E^0_{CO_2/CO}$ potential (Fig. 3D). Analogous to **C1**, the respective **C2**-driven formation of CO and $CO_2$ during the reductive and oxidative scans in aqueous media was confirmed by complementary GC data (Figs. S32, 33). The onset potential for both $CO_2$ reduction and CO oxidation for **C2** in water coincides with $E^0_{CO_2/CO}$ potential as observed during the lower scan rate data collected for the same sample under a 1:1 $CO_2$/CO atmosphere (Fig. 3F). This particular feature signifies the attainment of reversible $CO_2$ reduction/CO oxidation by **C2** in an aqueous media. Hence, the $CO_2$/CO reversibility, especially the CO oxidation side, can be regulated by the actual experimental condition, primarily due to its inherent bias to $CO_2$ reduction. The background corrected data for **C2** under 1:1 $CO/CO_2$ mixture in aqueous media also corroborated this conclusion (Fig. S34).

A range of rinse test experiments was performed for **C1** and **C2** to establish the homogeneous nature of the electrocatalytic responses in DMF (Fig. S35). The surface assessment of the working electrode via SEM study post-bulk electrolysis also confirmed no significant

formation of any copper-based heterogeneous material (Fig. S36). Additionally, the optical spectra of the complexes were recorded pre- and post-electrolysis for **C1** and **C2**. The similar comparative optical spectra for the complexes highlighted the stability of the complex under catalytic conditions (Fig. S37).

The electrocatalytic $CO_2$ reduction and CO oxidation rates for these complexes are calculated from the catalytic and stoichiometric current ratio using Equation S1. Although both **C1** and **C2** exhibited energy-efficient $CO_2$ reduction catalysis in the DMF medium, albeit at different rates. **C2** exhibits ~3.5 times faster $CO_2$ reduction (TOF $85 \pm 5\,s^{-1}$) compared to **C1** (TOF $24 \pm 3\,s^{-1}$). The presence of the amine functionality in **C2** and its potential involvement in rapid proton exchange apparently enhances $CO_2$ reduction catalysis. **C1** displays a nearly reversible catalytic behavior as it oxidizes CO close to $E^0_{CO_2/CO}$ with a rate (TOF $14 \pm 1\,s^{-1}$) similar to its $CO_2$ reduction, highlighting an unbiased behavior during the reversible catalysis. On the other hand, **C2** exhibited a distinctively slow CO oxidation rate (TOF $3 \pm 1\,s^{-1}$) at a distant potential from $E^0_{CO_2/CO}$. This data demonstrated a biased and bidirectional CO2RR catalytic behavior for **C2**. The catalytic performance for both catalysts improved in an aqueous medium while they attained catalytic reversibility for $CO_2$ reduction/CO oxidation. Under a 100% $CO_2$ atmosphere, **C1** catalyzes $CO_2$ reduction and CO oxidation at a rate of 25000 s⁻¹ and 4600 s⁻¹, respectively. On the other hand, **C2** exhibited an enhanced $CO_2$ reduction rate of 42600 s⁻¹, while its CO oxidation remained subdued (~100 s⁻¹) due to its inherent bias for $CO_2$ reduction. The improved $CO_2$ reduction by **C2** compared to **C1** is analogous to their performance in organic media, which can be attributed to the influence of the peripheral amine groups.

## Chemical catalysis

The simultaneous electrocatalytic $CO_2$ reduction and CO oxidation by **C1** and **C2** indicates that these complexes can instigate spontaneous $CO_2$ reduction via chemical catalysis. $CO_2$ was gradually purged into a violet-colored Cu(I) sample of **C1** in DMF to probe this hypothesis, which triggered a visible change as the solution turned yellow. Such a variation is primarily originated from the loss of characteristic Cu(I)-originated MLCT band (Fig. 5A). Interestingly, the changes in the optical spectrum were replicated with the chemical oxidation of **C1** by CAN in a parallel experiment (Fig. 5B). This observation indicated that $CO_2$ addition prompts Cu(I/II) oxidation at the **C1** core. The $CO_2$

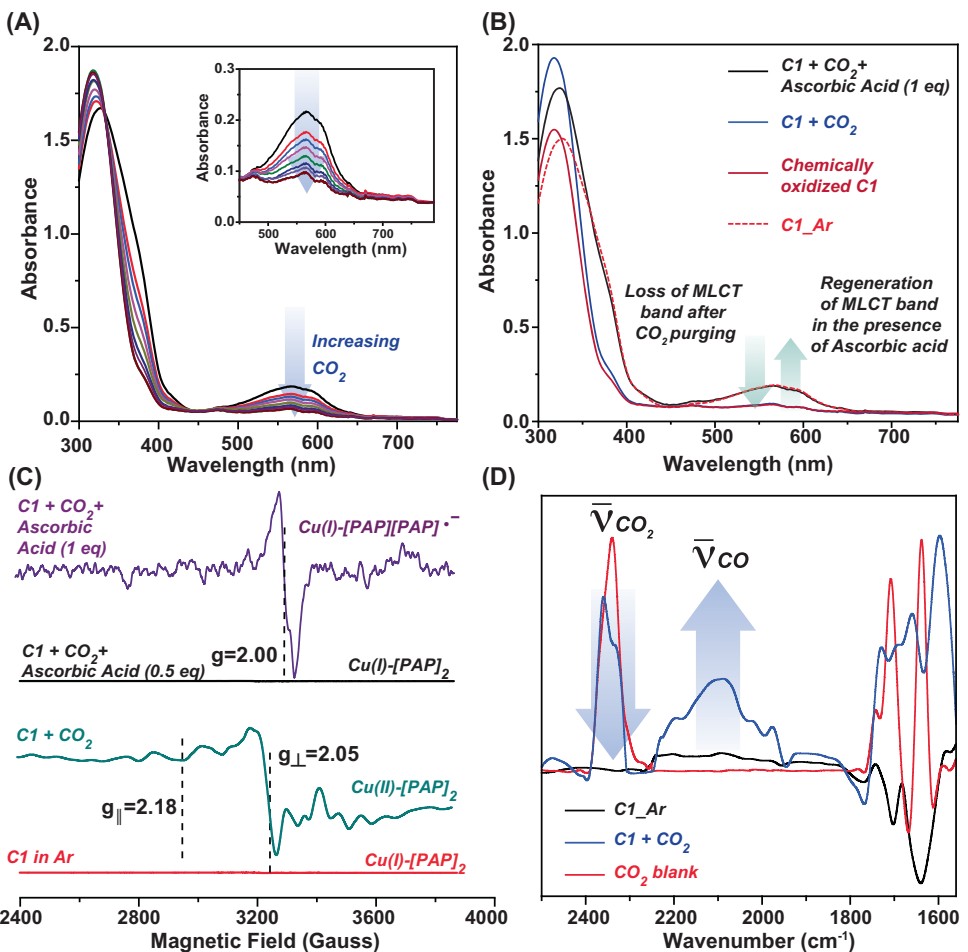

**Fig. 5 | Spectroscopic changes during chemical CO₂ reduction/CO oxidation interconversion. A** The sequential alteration in the optical spectra of **C1** in Ar-saturated DMF following gradual purging of CO₂. The inset specifically demonstrates the change in the MLCT band. **B** The serial changes in the optical spectra of native **C1** sample under Ar atmosphere (red dotted trace) following the purging of CO₂ (blue trace), and addition of one equivalent of ascorbic acid (black trace) along with a chemically (CAN) oxidized **C1** (brown trace). **C** The change in the EPR spectra of **C1** present in Ar (red trace) following the successive addition of CO₂ (sea green trace), 0.5 equivalent (black trace) and one equivalent of ascorbic acid (violet trace). **D** The comparative solution state FTIR spectra of **C1** under Ar (black trace) and CO₂ atmosphere (blue trace) along with a blank solution under CO₂ (red trace).

purged sample was paramagnetic, and it exhibited a signal that resembled Cu(II) signature (Fig. 5C, S6). This observation further strengthens the hypothesis that Cu(I) center is oxidized to Cu(II) following its interaction with CO₂. As CO₂ to CO conversion requires two electrons, it can be assumed that the redox-active (Phenylazo)pyridine (PAP) ligand also participates during this reaction by providing one electron along with the Cu center. To probe the involvement of the PAP ligand in the catalysis, a naturally abundant and widely employed chemical reductant ascorbic acid was added sequentially to the CO₂-purged **C1** sample (containing Cu(II)-[PAP]₂ resting state). First, 0.5 equivalent of ascorbic acid was added, which transfers one electron to the complex, and resulted in an EPR silent signal. This data was identical to the precursor **C1** complex, and it is attributed to Cu(I)-[PAP]₂ species. Next, another 0.5 equivalent ascorbic acid was added to include one more electron to the complex. Interestingly, a free radical EPR signal was observed in the resultant solution. This species is probably generated due to the formation of Cu(I)-[PAP][PAP]·⁻ species, where only one of the coordinating PAP ligands is reduced (Fig. 5C). Thus, these successive EPR experiments suggest that both the Cu(I) and redox-active PAP ligand supply one electron each during the two-electron CO₂/CO reduction. The sequential origination of Cu(I) followed by [PAP]·⁻ during the reduction of the precursor Cu(I)-PAP was also observed during the electrochemical studies (Fig. 2A).

The change in CO₂ treated **C1** solution was also followed by FTIR spectroscopy. The addition of CO₂ to **C1** in a DMF solution resulted in the appearance of new signals at 1750 cm⁻¹ that can be assigned to metal complex-bound CO₂ species (Fig. 5D)³⁸. The FTIR spectra also showcased a broad signal ~2100 cm⁻¹, which possibly demonstrates the formation of CO at the end of the catalytic cycle. The characteristic asymmetric stretching band of CO₂ ~2350 cm⁻¹ gradually disappeared, simultaneous to the emergence of the CO vibrational band (Fig. 5D). This observation corroborates further the **C1**-catalyzed conversion of CO₂ to CO. **C2** also followed a similar catalytic pathway under 1 atm CO₂ in the presence of ascorbic acid that was evident from the corresponding optical and FTIR spectra (Figs. S38, S39).

Next, the CO₂ reduction cycle was continued in the presence of multiple equivalents of ascorbic acid, and the active catalysis was evident from the appearance of CO as the primary CO2RR product via GC (Fig. S40). **C2** also exhibits similar chemical catalysis, albeit faster than **C1**, as observed even during the electrocatalytic studies. The complementary GC experiment displayed a turnover number (TON) of 6.0 for **C2** compared to 4.0 for **C1**, in the presence of ten equivalents of ascorbic acid in DMF solution under one atmospheric CO₂ over 22 hours (Fig. S41). **C1** was found to be impressively stable in DMF solution as it retains the chemical and electrocatalytic CO2RR activity over 30 days (Fig. S42).

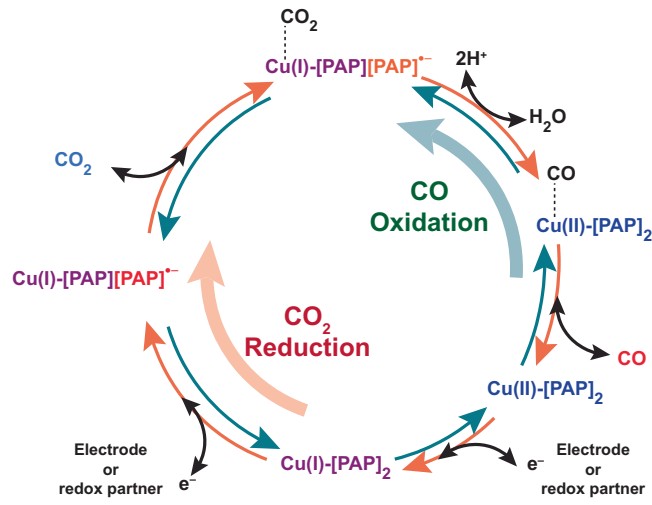

**Fig. 6 | The proposed catalytic cycle.** The possible catalytic mechanism for redox-active ligand coordinated copper complex (**C1** and **C2**)-mediated reversible $CO_2$ reduction (orange arrows) and CO oxidation (cyan arrows).

The other side of the reversible $CO_2$/CO transformation, i.e., CO oxidation, was also followed under chemical catalytic conditions for these complexes. It was noticed that interaction between CO and these complexes occurs once the central copper attains the +2-oxidation state. Hence, complexes **C1** and **C2** were chemically oxidized with CAN prior to their treatment with CO. The presence of water is vital in this step as it possibly supports the formation of a new C-O bond while promoting the redox change at the copper center[39]. The generation of $CO_2$ from a CO saturated DMF solution of **C1** or **C2** in the presence of 11% water (6.1 M) was again validated via GC data (Figs. S43, S44). When CO was added to a yellow-colored Cu(II) sample in DMF media, it spontaneously converted into a violet Cu(I) sample. This redox change is possibly correlated with CO oxidation to $CO_2$ as this two-electron process proceeds via Cu(I)-[PAP][PAP]$^{•-}$ intermediate. The serial addition of electron acceptor CAN restored the precursor Cu(II)[PAP]$_2$ species to trigger the next cycle of CO oxidation that was nicely depicted by the change in the optical spectra (Fig. S45).

All these data imply that the reversible catalytic pathway proceeds through a common pathway that possibly involves a Cu(I)-[PAP][PAP]$^{•-}$–$CO_2$ adduct. It can be generated either during a reaction between Cu(I) and $CO_2$ ($CO_2$ reduction) or during Cu(II)/CO interaction (CO oxidation). During $CO_2$ reduction, the catalytic cycle is further driven by sacrificial electron donor ascorbic acid that ensures the regeneration of Cu(I) at the end of the cycle (Fig. 6). Here, the protic ascorbic acid possibly supports the PCET-induced C-O bond cleavage and formation of CO and $H_2O$ molecules. This step can be further accelerated with the peripheral amine functionalities, which is supported by the relatively faster $CO_2$ reduction rate by **C2** as compared to **C1**. In the opposite direction, water molecules promote the interaction between Cu(II) and CO, leading to the formation of reduced Cu(I)-[PAP][PAP]$^{•-}$ species and $CO_2$ after a two-electron exchange process. Hence, the strongly π-accepting PAP ligand plays a vital role during the reversible $CO_2 \rightleftharpoons CO$ conversion by accommodating the need of the second electron by synchronously employing [PAP]/[PAP]$^{•-}$ and Cu(II/I) redox couples. Comparative FTIR spectra were recorded for **C1** under electrocatalytic conditions in the presence of 100% $CO_2$ atmosphere, where no significant change was observed at the azo-stretching band of the ligand (Fig. S46). This data indicates that the $CO_2$ binding is possibly triggered by the Cu center rather than the ligand environment. Later, the electrocatalytic CO2RR responses from the complexes under pure $CO_2$ were compared to $CO_2$/CO blended atmosphere. The onset potential of $CO_2$ reduction shifted to the anodic direction when

the local $CO_2$ concentration was lowered with CO mixing for both **C1** and **C2**, highlighting the reversible hallmark of the catalysis (Fig. S47).

## Discussion

In this study, we have probed the electrocatalytic behavior of copper-based homogeneous molecular catalysts that showcased impressive $CO_2$ reduction behavior whose onset potential values align with the $CO_2$/CO equilibrium potential. This observation highlights an energy-efficient $CO_2$ reduction persisted in both organic and aqueous media. Here, the presence of a redox-active ligand partner turns out to be a decisive factor behind this exceptional catalytic reactivity. Each π-interacting (phenylazo)pyridine (PAP) motif and the central copper provide the two electrons required for $CO_2$ to CO reduction during the electrocatalytic pathway. The generation of the vital intermediate species Cu(I)-[PAP][PAP]$^{•-}$ was spectroscopically confirmed during the reversible $CO_2$ reduction/CO oxidation cycle, respectively, deploying compatible sacrificial electron donor or acceptor molecules.

Among the two complexes, **C2** catalyzes $CO_2$ reduction faster compared to **C1**. The presence of the peripheral amine functionality in **C2** possibly plays a key role in displaying an enhanced $CO_2$ reduction behavior via PCET pathway. Interestingly, **C1** and **C2** also exhibit electrocatalytic CO oxidation during the returning anodic scan. However, the CO oxidation propensity varies between **C1** and **C2**. **C1** epitomizes an efficient catalyst where it triggers both $CO_2$ reduction and CO oxidation very close to the $CO_2$/CO thermodynamic potential in both organic and aqueous media. The catalytic rates for $CO_2$ reduction and CO oxidation also remained similar for **C1**, which indicates an unbiased reversible $CO_2$ reduction/CO oxidation behavior by **C1**.

On the other hand, **C2** demonstrated a bidirectional behavior in organic media where CO oxidation remained significantly subdued while its catalytic maximum remained remotely positioned from the $CO_2$/CO equilibrium potential. The inclusion of water in the solution improved the situation as CO oxidation onset potential shifts towards the $CO_2$/CO equilibrium potential along with an enhancement in $CO_2$ to CO catalysis. Finally, **C2** achieves a reversible $CO_2$ reduction/CO oxidation signature in 100% aqueous media under a 1:1 $CO_2$/CO atmosphere. Again, the presence of the amine groups in the periphery of **C2** is found to be influential in this alteration in biased electrocatalytic behavior.

Here, we have probed a unique way of replicating the metalloenzyme infrastructure into a synthetic functional molecular complex. Here, the rational inclusion of a multi-functional redox-active ligand framework around a copper core functionally imitates the protein-scaffolded iron-sulphur cluster present in [NiFe]-CODH active site. The resultant molecular catalysts exhibit not only $CO_2$ reduction reactivity but also achieve reversible catalytic behavior under regulated conditions, which is a hallmark of natural enzymes[30]. The energy-efficient catalytic response coupled with long-term stability can establish this first-row transition metal and redox-active ligand combination as a template for the development of industrial-scale catalytic materials. The efficient $CO_2$/CO conversion can readily minimize the carbon footprint in the steel industry by curbing the coke usage in a blast furnace[40]. On the other hand, this reversible $CO_2 \rightleftharpoons CO$ conversion can be integrated with the Fischer-Tropsch process along with green $H_2$ to create a close-looped carbon cycle with minimal environmental consequences[7]. Thus, introducing this energy-efficient, robust, and economic catalyst can actively revert atmospheric $CO_2$ to the carbon biogeochemical cycle to positively impact our pursuit of a carbon-neutral energy economy[41,42].

## Methods
### Synthetic Procedure
**2-(phenylazo)pyridine (L1).** The ligand 2-(phenyazo)pyridine (**L1**) was synthesized as per reported by Lahiri and co-workers[43]. In summary, 1.00 g 2-aminopyridine (10.60 mmol.) was added to a hot (60 °C) 50% aqueous NaOH solution (25 mL), followed by the addition of 3 mL of

benzene. Next, 1.20 g of nitrosobenzene (11.20 mmol), dissolved in 10 mL of benzene, was added by a dropping funnel to the previous mixture over 30 minutes and warmed for 45 minutes. After extraction with benzene (3 × 100 mL), the organic solution was refluxed with charcoal for 2 hours, filtered, and concentrated under reduced pressure. The final product was separated by column chromatography, using alumina as stationary phase and 1-5% DCM-Hexane as mobile phase. Yield 1.5 g. (77.3%).

**Bis-(2-(phenylazo)pyridine) copper(I) perchlorate (C1). C1** was synthesized following a modified version of the procedure reported by Datta et al. [27]. Here, 974.5 mg of 2-(phenylazo)pyridine (**L1**) (5.22 mmol) was added dropwise to a methanol/dichloromethane (1:2) blended solution (50 ml) containing 986.3 mg of $Cu(ClO_4)_2,6H_2O$ (2.66 mmol). The mixture was stirred overnight at room temperature, where a violet solution was obtained. This solution was evaporated under reduced pressure, and the residual solid was repetitively washed with n-pentane until the filtrate became colourless. Next, the violet colour sticky solid product was boiled in a 72 ml methanol-water mixture (3:1). The resulting precipitate was filtered and dissolved in methanol for crystallization. Violet colour crystals appeared after 1 week that was further dried under vacuum. Yield: 1.13 g (80.3% with respect to **L1**). HRMS (ESI, +ve mode, MeOH) m/z for (M⁺) $[C_{22}H_{18}Cu_1N_6]$: Calculated: 429.0883, Experimental: 429.0883 (Fig. S48A). UV-Vis in DMF ($\lambda_{max}$ in nm, $\varepsilon$ in parentheses $M^1 cm^{-1}$): $\lambda_{max}$ = 360 (25000); 580 (3929); 700 (907).

**6-amino-2(phenylazo)pyridine (L2).** The ligand 6-amino-2(phenylazo) pyridine (**L2**) was synthesized from 2,6-diaminopyridine as follows. 2.00 g of 2,6-diaminopyridine (18.33 mmol) was dissolved in 15.0 mL of pyridine and mixed with 10.0 mL of 60% aqueous NaOH solution. Then 1.95 g of nitrosobenzene (18.2 mmol), dissolved in 10.0 mL pyridine, was added dropwise to the mixture for 3.0 hours, followed by 20.0 hours of reflux. The completion of the reaction was confirmed by thin-layer chromatography. The dark red mixture solution was diluted with water and extracted with dichloromethane (DCM) (3 × 100 mL). A dark red crude was obtained from evaporation of the organic layer, which was purified by column chromatography (100% DCM), using neutral alumina as stationary phase. Yield 378.0 mg (10.4% with respect to 2,6-diaminopyridine).

$^1H$ NMR (400 MHz, δ in ppm, 298 K, $CDCl_3$); δ = 7.99 (d, J = 7.9 Hz, 1H); 7.60 (t, J = 7.8 Hz, 1H); 7.49 (d, J = 7.49 Hz, 3H); 7.18 (d, J = 7.5 Hz, 1H); 6.59 (d, J = 8.1 Hz, 1H); 4.90 (s, 2H) (Fig. S49).

**Bis-(6-amino-2(phenylazo)pyridine) copper(I) Bromide (C2).** The 6-amino-2-(phenylazo)pyridine (**L2**) (42.8 mg/0.22 mmol) in 10 ml DCM was dropwise added (under $N_2$) to a methanol/dichloromethane (1:2) mixture containing 40.0 mg of $Cu(ClO_4)_2,6H_2O$ (0.11 mmol). The mixture solution was stirred for 1 hour at room temperature. Finally, a green solution was obtained that was evaporated under the vacuum. Then the green solid remaining was boiled in a methanol-water mixture. The precipitate from the mixture was filtered and dissolved in methanol and recrystallized in methanol with diffusing n-hexane. Small green crystals appeared in a week that was dried under vacuum. Yield 15.0 mg (25% with respect to **L2**). HRMS (ESI, +ve mode, MeOH) m/z for (M⁺) $[C_{22}H_{20}Cu_1N_8]$: Calculated: 459.1101, Experimental: 459.1101 (Fig. S48B). UV-Vis in DMF ($\lambda_{max}$ in nm, $\varepsilon$ in parentheses $M^1 cm^{-1}$): $\lambda_{max}$ = 425 (26500); 601 (5860); 739 (1233).

**Cyclic voltammetry study in organic solvent and in aqueous pH buffer.** The cyclic voltammetry experiment was primarily executed in an organic medium (dry DMF). Additionally, a few experiments were performed in an aqueous solution (0.5 M $NaHCO_3$ buffered medium, pH 6.5). The analyte complex concentration was maintained at ~1 mM in all the cases unless mentioned otherwise. In an organic medium,

0.1 M tetrabutylammonium tetrafluoroborate (TBAF, $nBu_4N^+BF_4^-$) was employed as the supporting electrolyte. In water, $Na_2SO_4$ was added for the same role. The cyclic voltammograms were recorded using a typical three-electrode assembly, containing a 1 mm diameter glassy carbon disc working electrode, Pt-wire counter electrode, and Ag/AgCl (saturated KCl) reference electrodes. The applied potential values during the experiments were internally standardized either by using ferrocene (in organic medium) or hydroxymethyl ferrocene (FcOH). Hence, all the potential values in organic media in this study were reported against ferrocene couple ($Fc^{+/0}$), while the same in aqueous media was done against RHE by referencing hydroxymethyl ferrocene couple ($FcOH^{+/0}$ ~ +0.385 V).

**Bulk or control potential experiment in aqueous buffer.** Bulk Electrolysis (BE) or control potential experiment (CPE) was performed in an air-tight 95 ml four neck glass vessel. Three of these outlets were fitted with various electrodes; 2 cm × 1 cm vitreous carbon as a working electrode; 23 cm coiled platinum wire as a counter electrode, and Ag wire as a reference electrode. The last outlet was closed by a B-14/20 suba® seal rubber septum, which was used for purging $CO_2$ or CO or Ar (for 30 minutes) before the experiments and for headspace gas collection. During an experiment, 14 ml of 1 mM complexes were added to the vessel, all electrodes (along with a magnetic bead) were inserted along with a B-14/20 rubber septum cap (in a gas-tight manner). Then, the chrono-coulometric experiment was started at corresponding catalytic potentials in both organic and aqueous media. The reaction solution was continuously stirred during the experiment. Headspace gas was collected by a GASTIGHT® PTFE leur-lock 1000 series (1001TLL) 1 ml Hamilton® syringe after certain time intervals, and it was analyzed via gas chromatography (GC) instrument on TCD/FID mode.

**Spectroelectrochemistry (optical).** Optical spectroelectrochemistry experiments were performed via an Ocean Optics spectrophotometer in tandem with a Metrohm Autolab PGSTAT204 potentiostat. The sample was placed in a 3.5 ml quartz cuvette (1 cm path length) fixed in an external sample holder and connected to a light source and a detector via optical fibres. The cuvette was also fitted with a 3 mm glassy carbon rod working electrode, a Pt wire counter electrode, and a silver wire reference electrode. A controlled potential electrolysis (CPE) experiment was performed with the use of the potentiostat, while the respective changes in the optical spectrum were monitored with the spectrophotometer.

**Spectroelectrochemistry (FTIR).** An optically transparent thin–layer electrode (OTTLE) cell, equipped with a Pt mesh working electrode, a Pt microwire counter electrode, and Ag microwire pseudo–reference electrode were employed for this study along with a Metrohm Autolab PGSTAT204. All the FTIR spectra were measured using a Perkin–Elmer FTIR spectrometer set in absorbance mode. All the measurements were carried out at room temperature with a ~ 30 mM complex concentration. Data was also recorded for a $CO_2$-saturated blank DMF solution, subtracted from all the experimental data collected under the $CO_2$ atmosphere.

**Rinse test.** A rinse test for complexes has been carried out in an organic medium (DMF) to probe the homogeneous or heterogeneous nature of the catalysis. For this purpose, we have executed three consecutive runs as follows. A complete CV was recorded for the complexes in the corresponding organic medium under the $CO_2$ atmosphere in the first run. Then the working electrode was thoroughly rinsed with water. Then this electrode was properly polished with 0.25 μm alumina powder. Afterward, a second run was performed with the same complex solution with the cleaned working electrode. However, this second run was stopped at the potential where the

maximum $CO_2$ reduction signal was observed. Then working electrode is only rinsed (without any polishing) with water, and a third cyclic voltammogram was recorded in a different solution that contained only blank DMF and the electrolyte but no complex. This third scan was initiated again from the maximum catalytic response potential observed for $CO_2$ reduction. The absence of any significant catalytic response in the third scan in the reduction direction validates the homogeneous $CO_2$ reduction mechanism.

**Chemical catalysis.** ~1 mM stock solutions for each complex were prepared in DMF. From that stock solution, 5 mL was taken in a 105 mL Schlenk tube connected to a magnetic rotor. Next, a stream of $CO_2$ was purged into the solution for 30 minutes along with stirring to prepare a $CO_2$-saturated solution. Then varying amounts of (0-10 mM) ascorbic acid were added to this Schlenk tube under anaerobic conditions while stirring at 500 rpm for 22 h. During this experiment, 0.5 ml headspace gas was taken out from the Schlenk tube and injected into the GC for analysis at various time intervals.

A similar procedure was adopted for CO oxidation with the sample under CO-saturated conditions. Here, cerium ammonium nitrate (CAN) and 6.1 M water were added instead of ascorbic acid to regenerate Cu(II) species. A number of control experiments were performed in the presence of ascorbic acid or CAN under $CO_2$ and CO atmosphere, respectively, in blank DMF.

**Gas chromatography analysis.** The amount of $CO_2$/CO evolved during catalysis were quantified by using Dhruva CIC gas chromatography (GC) instrument with TCD/FID detector with a 5 Å molecular sieve/Porapak at room temperature. Instrument calibration build up curve was created manually by injecting variable amount (0.5%-2%) of known $CO_2$ and CO gas mixtures.

**Calculation of catalytic rate.** Catalytic rate of the complexes was measured from this Eq. (1):

$$\frac{i_{cat}}{i_p} = \frac{n}{0.4463}\sqrt{\frac{RTk_{obs}}{F\nu}} \tag{1}$$

where, $i_{cat}$ = catalytic current, $i_p$ = stoichiometric current, n = number of electrons involved in this process, R = universal gas constant, T = temperature in K, F = 1 Faraday, and $\nu$ = scan rate.

## Data availability
The data that support the findings of this study are available within the paper and its Supporting Information file.

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

## Acknowledgements

The authors would like to acknowledge the support from DST, India-supported National Center of Excellence in Carbon Capture and Utilization (DST/TMD/CCUS/CoE/2O2/IITB), SERB, India (IMP/2018/001208), and CSIR (01(3037)/21/EMR-II) for this research activity. The authors would also like to thank IIT Bombay for the research facilities. The authors also acknowledge Prof. Jyotishman Dasgupta, TIFR Mumbai, for his support with EPR experiments.

## Author contributions

So.G. and A.D. conceptualized the project. So.G. (Somnath) synthetized, characterized, and performed the major experiments. D.D., C.D., and Sa.G. performed the control electrochemical experiments for $CO_2$ reduction and CO oxidation. So.G., D.D., V.V., D.M., G.K.L., and A.D. analyzed the data. So.G. and A.D. wrote the first draft of the manuscript. So.G., D.D., V.V., D.M., G.K.L., and A.D. revised the manuscript. AD directed the project. All the authors participated in evaluating the results and commented on the manuscript.

## Competing interests

So.G., D.D., C.D., Sa.G., V.V., D.M., G.K.L., and A.D. have been granted a patent based on this work along with IIT Bombay (India Patent No. 435947, Indian Patent Application No: 202221011195).
