## [Peer Review File · Nature Communications]

REVIEWER COMMENTS

Reviewer #1 (Remarks to the Author):

The authors have significantly improved the manuscript on their interesting findings on bidirectional/reversible catalysis for CO₂/CO interconversion. For publication in Nature Communication, I would still recommend a more quantitative description of the catalytic behavior of their catalysts:

1. Generally the authors assign catalytic performance of their "Complex 1" as reversible while it is not strictly the case for all data described as such. For instance they state: „We have assigned these signals as CO₂ reduction and CO oxidation features, respectively, where the catalysis occurs reversibly with minimal overpotential requirements in either direction.“ However the data shown in figure 1A display a clear change in slope when transitioning from CO₂ reduction to CO oxidation showing that catalysis remains bidirectional. Even if the overpotential is low, it is not minimal. The wording “nearly reversible” or better “bidirectional” should be used instead of “reversible for all measurements in organic solvents. On the other hand catalysis in pure aqueous solvents seem in deed very close to reversible as highlighted by the measurements under CO/CO₂ (Figure 3E). Adjusting the experimental conditions to avoid mass transport limitation could make this highly convincing (see following point).

2. Generally, the paper lacks quantification of the catalytical potentials for both CO oxidation and CO₂ reduction which is complicated by mass transport limitations of the substrate in all catalytic currents (peak shaped currents) shown in the manuscript. Adjusting the catalyst concentration (using less catalyst) vs the substrate concentration under CO/CO₂ atmosphere, and adjusting the scan rate of the measurement could help reach a catalytic regime limited by catalysis rather than diffusion to enable reliable quantification of the catalytic properties. The author could check how this was carried out for a reversible catalysts for H⁺/H₂ interconversion (Figure 2 and 3 in Angew. Chem. Int. Ed.2023, e202302779). The data in Figure 3E and 3F in the present manuscript could be good starting points for this (for instance reducing catalyst concentrations at least 10 fold may allow to reach a regime where the catalytic current is limited by catalysis only). This would be highly valuable to support the conclusion of the present manuscript and to compare the catalytic performance with other previously reported reversible/bidirectional catalysts for CO₂ reduction or H⁺/H₂ interconversion.

Reviewer #2 (Remarks to the Author):

In this work, a copper center combining redox-active ligands unveiled a reversible CO₂ reduction and CO oxidation. The inclusion of amine groups around the copper center provides biases toward CO₂ reduction in organic/aqueous media. However, I found many are missing at this shape.

1. The copper is saturated in coordination and how could the copper center perform catalysis? However, Cu may leach out to deposit as Cu(0), then there is chance to perform catalysis.

2. Lack of electrolysis data. No product/potential dependence is performed.

3. Page 6 & Figure 2c

"The Cu(I) signature absorbance band ~580 nm disappeared following an applied potential in the anodic direction (-0.14 V vs. FeCp₂^{+/0}), which re-emerged once a reductive potential (-0.35 V vs. FeCp₂^{+/0}) was implemented (Figure 2C, Figure S6). Thus, this signal is ascribed as a Cu(II/I) reduction."

There is a confusion for the legends of "C1 BE@-0.35V" and "C1 BE@-0.14V" in Figure 2C.

4. The amount of CO produced by CO₂ reduction during the CV is usually very few, is it enough to generate CO oxidation signals in CV? It is necessary to make a control experiment in CO atmosphere.

5. Figure S27.

It is unusual to observe C₂H₄ signal in GC. In order to recognize the source of C₂H₄ (product of CO₂ reduction or impurity), it is necessary to make a contrast experiment in Ar or N₂ atmosphere.

4. Page 15

"The presence of water is vital in this step as it possibly supports the formation of a new C-O bond while promoting the redox change at the copper center."

It needs citation of references to support this statement.

Reviewer #3 (Remarks to the Author):

In this manuscript, the authors mimicked the carbon monoxide dehydrogenase enzyme by combining redox-active ligands around a copper center. This approach allows for reversible CO₂ reduction/CO oxidation catalysis, similar to natural enzymes. By including proton-exchanging amine groups in the copper complex, the pathway for CO₂ reduction and CO oxidation could be adjusted in both organic and aqueous environments.

In the revision, the authors have pretty much solved the concerns raised by previous reviewers. But I do not think Figure 3A and Figure S12A are the kind of Faradaic efficiency that previous reviewer 3 asked. Another revision would be needed to provide this information.

Dr. Arnab Dutta
Associate Professor
Chemistry Department
Associate
Interdisciplinary Programme in
Climate Studies
IIT Bombay
Maharashtra India 400076

Email: arnabdutta@chem.iitb.ac.in
arnab.dutta@iitb.ac.in
Contact: +91-9537995998

REVIEWER COMMENTS

Reviewer #1 (Remarks to the Author):

The authors have significantly improved the manuscript on their interesting findings on bidirectional/reversible catalysis for CO₂/CO interconversion. For publication in Nature Communication, I would still recommend a more quantitative description of the catalytic behavior of their catalysts:

1. Generally the authors assign catalytic performance of their "Complex 1" as reversible while it is not strictly the case for all data described as such. For instance they state: „We have assigned these signals as CO₂ reduction and CO oxidation features, respectively, where the catalysis occurs reversibly with minimal overpotential requirements in either direction.“ However the data shown in figure 1A display a clear change in slope when transitioning from CO₂ reduction to CO oxidation showing that catalysis remains bidirectional. Even if the overpotential is low, it is not minimal. The wording “nearly reversible” or better “bidirectional” should be used instead of “reversible for all measurements in organic solvents. On the other hand catalysis in pure aqueous solvents seem in deed very close to reversible as highlighted by the measurements under CO/CO₂ (Figure 3E). Adjusting the experimental conditions to avoid mass transport limitation could make this highly convincing (see following point).

Response: We appreciate the excellent suggestion from the reviewer. We have now probed the catalytic behaviour of the **complex C1** in detail (in organic media) to have a better insight into its CO₂ reduction and CO oxidation signals and analyzed the electrochemical signals in the context of reversibility. In this regard, we have prepared different concentration variants of the complex in DMF and recorded its response under a 1:1 CO/CO₂ gas mixture. Here, we have recorded the cathodic CO₂ reduction signal and anodic CO oxidation signal in a single run. The scan started at the equilibrium potential, and the initial scan direction was at the anodic direction. Here, the background current response was also collected and duly subtracted from the obtained catalytic data beyond the potential where the catalytic current response was noticed. As shown in the following figure, the current response continued to grow beyond the equilibrium potential in either redox direction.

Dr. Arnab Dutta
Associate Professor
Chemistry Department
Associate
Interdisciplinary Programme in
Climate Studies
IIT Bombay
Maharashtra India 400076

Email: arnabdutta@chem.iitb.ac.in
arnab.dutta@iitb.ac.in
Contact: +91-9537995998

This procedure was applied for the data recorded for **C1** in 1:1 CO/CO₂ in DMF as displayed below.

This data was recorded for complex **C1** at variable concentrations (**Fig. A**). Here, we have observed that the catalytic CO₂ reduction and CO oxidation signals are directly dependent on catalyst concentration. At lower concentrations, the current responses remain low; however, the distinct signature of both CO₂ reduction and CO oxidation are visible on either side of the equilibrium potential (**Insert of Figure A**). Here, the scan rate was kept constant (50 mVs⁻¹) for recording each data.

Dr. Arnab Dutta
Associate Professor
Chemistry Department
Associate
Interdisciplinary Programme in
Climate Studies
IIT Bombay
Maharashtra India 400076

Email: arnabdutta@chem.iitb.ac.in
arnab.dutta@iitb.ac.in
Contact: +91-9537995998

Next, a series of data was recorded for the same concentration of complex **C1** (1.0 mM) at a variable scan rate. Here, the current responses varied with altering scan rates. However, the background corrected data showcase the presence of both CO₂ reduction and CO oxidation at all conditions. At low scan rates, the catalyst showcases bias towards CO₂ reduction compared to CO oxidation.

The background corrected data for **C2** has now been added in the SI segment with appropriate discussion in the main text.

We have also mentioned the catalytic behaviour as “nearly reversible” in organic solvents as suggested by the reviewer.

2. Generally, the paper lacks quantification of the catalytical potentials for both CO oxidation and CO₂ reduction which is complicated by mass transport limitations of the substrate in all catalytic currents (peak shaped currents) shown in the manuscript. Adjusting the catalyst concentration (using less catalyst) vs the substrate concentration under CO/CO₂ atmosphere, and adjusting the scan rate of the measurement could help reach a catalytic regime limited by catalysis rather than diffusion to enable reliable quantification of the catalytic properties. The author could check how this was carried out for a reversible catalysts for H⁺/H₂ interconversion (Figure 2 and 3 in *Angew. Chem. Int. Ed.* 2023, e202302779). The data in Figure 3E and 3F in the present manuscript could be good starting points for this (for instance reducing catalyst concentrations at least 10 fold may allow to reach a regime where the

Dr. Arnab Dutta
Associate Professor
Chemistry Department
Associate
Interdisciplinary Programme in
Climate Studies
IIT Bombay
Maharashtra India 400076

Email: arnabdutta@chem.iitb.ac.in
arnab.dutta@iitb.ac.in
Contact: +91-9537995998

catalytic current is limited by catalysis only). This would be highly valuable to support the conclusion of the present manuscript and to compare the catalytic performance with other previously reported reversible/bidirectional catalysts for CO₂ reduction or H⁺/H₂ interconversion.

Response: We thank the reviewer for this valuable suggestion, and we have executed the experiment of **C1** under 1:1 CO₂/CO conditions at (A) variable concentrations (0.1, 0.2, and 1.0 mM) and (B) at variable scan rates (20, 50, and 100 mVs⁻¹). We have followed the lead provided by the authors in the article for probing *DuBois*-type catalyst for H₂ oxidation/H₂ production activity (*Angew. Chem. Int. Ed.*, 2023, e202302779, *Proc. Natl. Acad. Sci.*, 111, 16286–16291, 2014). We have now provided a similar treatment to probe the reversible vs. bidirectional nature (for CO₂ reduction/CO oxidation) of **C1** under variable conditions. As show in the figure below, **C1** demonstrated reversible CO₂ reduction/CO oxidation in DMF when 1:1 CO/CO₂ mixture was present.

Reviewer #2 (Remarks to the Author):

In this work, a copper center combining redox-active ligands unveiled a reversible CO_2 reduction and CO oxidation. The inclusion of amine groups around the copper center provides biases toward CO_2 reduction in organic/aqueous media. However, I found many are missing at this shape.

1. The copper is saturated in coordination and how could the copper center perform

Dr. Arnab Dutta
 Associate Professor
 Chemistry Department
 Associate
 Interdisciplinary Programme in
 Climate Studies
 IIT Bombay
 Maharashtra India 400076

Email: arnabdutta@chem.iitb.ac.in
arnab.dutta@iitb.ac.in
 Contact: +91-9537995998

catalysis? However, Cu may leach out to deposit as Cu(0), then there is chance to perform catalysis.

Response: We thank the reviewer for this excellent comment. We performed a series of experiments to probe the possible involvement of active Cu(0) particles in catalytic activity. Here, we initially performed the rinse test that showcased homogeneous nature of the catalysts. Later on, we executed bulk electrolysis of the both complexes and analyzed the working electrode's surface via SEM-EDS experiment. This experiment showcased no such formation of any copper nanoparticles (please see below), which further corroborated that the solvent-based copper complexes are responsible for the electrocatalytic behavior.

We also exhaustively investigated the reaction solution with optical spectroscopy after the bulk electrolysis that showcased no significant changes in the optical spectra to indicate the copper complexes remained stable during the electrocatalysis (Shown below).

Dr. Arnab Dutta
Associate Professor
Chemistry Department
Associate
Interdisciplinary Programme in
Climate Studies
IIT Bombay
Maharashtra India 400076

Email: arnabdutta@chem.iitb.ac.in
arnab.dutta@iitb.ac.in
Contact: +91-9537995998

The electron-rich low-valent Cu(I) center is prone to transfer its electron density to electrophilic CO₂. This hypothesis was supported by the rapid conversion of Cu(I) centre to Cu(II) in the presence of CO₂, which was followed via optical spectroscopy. Here the loss of the distinct MLCT band was the vital change observed during this experiment (following figure). It further suggests that CO₂ is binding with complexes at Cu(I) state. This change can be cycled chemically in the presence of sacrificial electron donor ascorbic acid.

We have now performed another extra set of FTIR-spectro-electrochemistry for complex **C1** in both CO₂ and Ar -based conditions. Here, we have specifically monitored the azo-stretching band before and after electrolysis. During this experiment, we found that the intensity and

Dr. Arnab Dutta
Associate Professor
Chemistry Department
Associate
Interdisciplinary Programme in
Climate Studies
IIT Bombay
Maharashtra India 400076

Email: arnabdutta@chem.iitb.ac.in
arnab.dutta@iitb.ac.in
Contact: +91-9537995998

position of the azo-stretching band remains almost identical under CO₂ atmosphere before and after electrolysis. These results also indicates that CO₂ binding is primarily occurring at the metal center while exerting minimal influence at the ligand side.

2. Lack of electrolysis data. No product/potential dependence is performed.

Response: We thank to the reviewer for this very important comment. We have now included all the Faradaic efficiency and product analysis data recorded for the all the electrolytic processes.

Dr. Arnab Dutta
Associate Professor
Chemistry Department
Associate
Interdisciplinary Programme in
Climate Studies
IIT Bombay
Maharashtra India 400076

Email: arnabdutta@chem.iitb.ac.in
arnab.dutta@iitb.ac.in
Contact: +91-9537995998

Figure. Bulk electrolysis of **C1** complex in CO₂ atmosphere. (A) Current vs. time plots at variable potentials. (B) Corresponding charge passed vs. time plots.

Figure. Bulk electrolysis of **C2** complex in CO₂ atmosphere. (A) Current vs. time plots at variable potentials. (B) Corresponding charge passed vs. time plots.

Figure. Bulk electrolysis of **C1** and **C2** complexes in CO atmosphere. (A) Current vs. time plots (B) Corresponding charge passed vs. time plots.

Dr. Arnab Dutta
Associate Professor
Chemistry Department
Associate
Interdisciplinary Programme in
Climate Studies
IIT Bombay
Maharashtra India 400076

Email: arnabdutta@chem.iitb.ac.in
arnab.dutta@iitb.ac.in
Contact: +91-9537995998

Figure. Comparative Faradic efficiency (FE) of **C1** and **C2** complexes for each bulk electrolysis experiment. (A) FE for CO formation by **C1** complex (B) FE for CO formation by **C2** complex. (C) FE for CO₂ formation by **C1** and **C2** complexes.

3. Page 6 & Figure 2c

"The Cu(I) signature absorbance band ~580 nm disappeared following an applied potential in the anodic direction (-0.14 V vs. FeCp₂^{2+/0}), which re-emerged once a reductive potential (-0.35 V vs. FeCp₂^{2+/0}) was implemented (Figure 2C, Figure S6). Thus, this signal is ascribed as a Cu(II/I) reduction."

There is a confusion for the legends of "C1 BE@-0.35V" and "C1 BE@-0.14V" in Figure 2C.

Response: We again thank the reviewer for the sharp observations. We revised all the details according to the new data in revised manuscript.

4. The amount of CO produced by CO₂ reduction during the CV is usually very few, is it enough to generate CO oxidation signals in CV? It is necessary to make a control experiment in CO atmosphere.

Dr. Arnab Dutta
Associate Professor
Chemistry Department
Associate
Interdisciplinary Programme in
Climate Studies
IIT Bombay
Maharashtra India 400076

Email: arnabdutta@chem.iitb.ac.in
arnab.dutta@iitb.ac.in
Contact: +91-9537995998

Response: We have now performed the experiment in 1:1 CO/CO₂ environment in organic media (DMF) where we were able to specifically probe the CO₂ reduction and CO oxidation activities.

5. Figure S27.

It is unusual to observe C₂H₄ signal in GC. In order to recognize the source of C₂H₄ (product of CO₂ reduction or impurity), it is necessary to make a contrast experiment in Ar or N₂ atmosphere.

Response: We again thank the reviewer for the observation. During the product isolation, we have recorded the GC spectrum of the pre-electrolysis gas sample and found the presence of

Dr. Arnab Dutta
Associate Professor
Chemistry Department
Associate
Interdisciplinary Programme in
Climate Studies
IIT Bombay
Maharashtra India 400076

Email: arnabdutta@chem.iitb.ac.in
arnab.dutta@iitb.ac.in
Contact: +91-9537995998

C_2H_4 as impurity in the CO_2 gas sample used during the catalysis. Here, the amount of C_2H_4 in pre- and post-electrolysed samples remained same.

4. Page 15

"The presence of water is vital in this step as it possibly supports the formation of a new C-O bond while promoting the redox change at the copper center."

It needs citation of references to support this statement.

Response: We acknowledge the reviewer for this very important suggestion. We have now included relevant citation (Ragsdale et al., Chem. Rev., 1996, 96, 2515-2540) in the revised manuscript.

Reviewer #3 (Remarks to the Author):

In this manuscript, the authors mimicked the carbon monoxide dehydrogenase enzyme by combining redox-active ligands around a copper center. This approach allows for reversible CO_2 reduction/ CO oxidation catalysis, similar to natural enzymes. By including proton-exchanging amine groups in the copper complex, the pathway for CO_2 reduction and CO

Dr. Arnab Dutta
Associate Professor
Chemistry Department
Associate
Interdisciplinary Programme in
Climate Studies
IIT Bombay
Maharashtra India 400076

Email: arnabdutta@chem.iitb.ac.in
arnab.dutta@iitb.ac.in
Contact: +91-9537995998

oxidation could be adjusted in both organic and aqueous environments.

In the revision, the authors have pretty much solved the concerns raised by previous reviewers. But I do not think Figure 3A and Figure S12A are the kind of Faradaic efficiency that previous reviewer 3 asked. Another revision would be needed to provide this information.

Response: We thank to the reviewer for this very important comment. We have now included all the Faradaic efficiency and product analysis data recorded for the all the electrolytic processes.

Figure. Bulk electrolysis of **C1** complex in CO₂ atmosphere. (A) Current vs. time plots at variable potentials. (B) Corresponding charge passed vs. time plots.

Dr. Arnab Dutta
Associate Professor
Chemistry Department
Associate
Interdisciplinary Programme in
Climate Studies
IIT Bombay
Maharashtra India 400076

Email: arnabdutta@chem.iitb.ac.in
arnab.dutta@iitb.ac.in
Contact: +91-9537995998

Figure. Bulk electrolysis of **C2** complex in CO₂ atmosphere. (A) Current vs. time plots at variable potentials. (B) Corresponding charge passed vs. time plots.

Figure. Bulk electrolysis of **C1** and **C2** complexes in CO atmosphere. (A) Current vs. time plots (B) Corresponding charge passed vs. time plots.

Dr. Arnab Dutta
Associate Professor
Chemistry Department
Associate
Interdisciplinary Programme in
Climate Studies
IIT Bombay
Maharashtra India 400076

Email: arnabdutta@chem.iitb.ac.in
arnab.dutta@iitb.ac.in
Contact: +91-9537995998

Figure. Comparative Faradic efficiency (FE) of **C1** and **C2** complexes for each bulk electrolysis experiment. (A) FE for CO formation by **C1** complex (B) FE for CO formation by **C2** complex. (C) FE for CO₂ formation by **C1** and **C2** complexes.

REVIEWERS' COMMENTS

Reviewer #1 (Remarks to the Author):

The revisions including the catalytic current under CO₂/CO feed at various catalyst concentrations and scan rates are convincingly demonstrating bidirectionality and near reversibility of this catalyst under these conditions (Figure S14). This strongly supports the conclusion and importance of the manuscript.

I would recommend to put figure S14 in the main text since it is much clearer than any of the other electrochemical data shown in the main text.

Minor comment: The inset in the figure is not described in the figure caption.

This data will be valuable to enable future quantitative analysis of this bidirectional/reversible catalyst.

I recommend publication of the manuscript in Nature Communications.

Reviewer #2 (Remarks to the Author):

ok to publish

Dr. Arnab Dutta
Associate Professor
Chemistry Department
Associate
Interdisciplinary Programme in
Climate Studies
IIT Bombay
Maharashtra India 400076

Email: arnabdutta@chem.iitb.ac.in
arnab.dutta@iitb.ac.in
Contact: +91-9537995998

REVIEWERS' COMMENTS

Reviewer #1 (Remarks to the Author):

The revisions including the catalytic current under CO₂/CO feed at various catalyst concentrations and scan rates are convincingly demonstrating bidirectionality and near reversibility of this catalyst under these conditions (Figure S14). This strongly supports the conclusion and importance of the manuscript. I would recommend to put figure S14 in the main text since it is much clearer than any of the other electrochemical data shown in the main text.

Response: We sincerely thank the reviewer for the excellent suggestion. We have now included the figure S14 in the main text accordingly.

Minor comment: The inset in the figure is not described in the figure caption.

Response: We thank the reviewer for this point. We have now modified the figure legend accordingly.

These data will be valuable to enable future quantitative analysis of this bidirectional/reversible catalyst.

I recommend publication of the manuscript in Nature Communications.

Response: We thank the reviewer for the excellent support.

Reviewer #2 (Remarks to the Author):

ok to publish

Response: We thank the reviewer for the excellent support.